# Optimized 18F-FDG PET-CT Method to Improve Accuracy of Diagnosis of Metastatic Cancer

**DOI:** 10.3390/diagnostics13091580

**Published:** 2023-04-28

**Authors:** Richard Black, Jelle Barentsz, David Howell, David G. Bostwick, Stephen B. Strum

**Affiliations:** 1AccuQuan, P.O. Box 89, Chagrin Falls, OH 44022, USA; 2Department of Radiology, Andros Clinics, Meester E.N. van Kleffensstraat 5, 6842 CV Arnhem, The Netherlands; j.barentsz@andros.nl; 3Department of Radiation Oncology, Ohio Health Cancer Center, 75 Hospital Drive, Athens, OH 45701, USA; xhowell@gmail.com; 4Rampart Health, 601 Biotech Drive, North Chesterfield, VA 23235, USA; dbostwick@ramparthealth.com; 5Community Practice of Hematology, Oncology and Internal Medicine, Focus on Prostate Cancer and Prostate Diseases, Medford, OR 97504, USA; sbstrum@gmail.com

**Keywords:** FDG PET, 18F PET-CT, glucose metabolism, standardized uptake value (SUV), cancer, nuclear medicine

## Abstract

The diagnosis of cancer by FDG PET-CT is often inaccurate owing to subjectivity of interpretation. We compared the accuracy of a novel normalized (standardized) method of interpretation with conventional non-normalized SUV. Patients (*n* = 393) with various malignancies were studied with FDG PET/CT to determine the presence or absence of cancer. Target lesions were assessed by two methods: (1) conventional SUV_max_ (conSUV_max_) and (2) a novel method that combined multiple factors to optimize SUV (optSUVmax), including the patient’s normal liver SUV_max_, a liver constant (k) derived from a review of the literature, and use of site-specific thresholds for malignancy. The two methods were compared to pathology findings in 154 patients being evaluated for mediastinal and/or hilar lymph node (MHLNs) metastases, 143 evaluated for extra-thoracic lymph node (ETLNs) metastases, and 96 evaluated for liver metastases. OptSUV_max_ was superior to conSUV_max_ for all patient groups. For MHLNs, sensitivity was 83.8% vs. 80.7% and specificity 88.7% vs. 9.6%, respectively; for ETLNs, sensitivity was 92.1% vs. 77.8% and specificity 80.1% vs. 27.6%, respectively; and for lesions in the liver parenchyma, sensitivity was 96.1% vs. 82.3% and specificity 88.8% vs. 23.0%, respectively. Optimized SUV_max_ increased diagnostic accuracy of FDG PET-CT for cancer when compared with conventional SUV_max_ interpretation.

## 1. Introduction

Modern imaging techniques enhance the accuracy of diagnosis, staging, and assessment of treatment outcome of various malignancies. In 1956, Warburg and associates [1] demonstrated that malignant cells preferentially utilize anaerobic consumption of glucose as their primary energy source, (i.e., “cancer’s sweet tooth”) [2]. Subsequent advances resulted in emergence of radiolabeled 18F fluorodeoxyglucose (FDG) positron emission tomography (PET) for physiologic imaging. PET is routinely combined with computed tomographic (CT) imaging for determination of anatomic detail [3,4] and attenuation correction [5,6,7,8].

PET-CT imaging in oncology focuses on differentiation of normal uptake, abnormal non-malignant uptake, and abnormal malignant uptake of the injected radiopharmaceutical [3]. One approach compared visual uptake in the target lesion with that in the mediastinal blood pool or liver parenchyma [9]. However, this qualitative approach was subjective and could be associated with poor reproducibility. Semi-quantitative evaluation is possible with 18 FDG PET through the generation of count statistics reflecting uptake in malignant target lesions, including the standardized uptake value (SUV) for differentiating malignant and non-malignant disease [10]. SUV measures the amount of radiolabeled glucose within a target lesion corrected for the administered dose of FDG and the patient’s body weight.

An SUV of 2.5 and above is a widely-used threshold to categorize a lesion as malignant [11], although multiple studies suggest different thresholds to identify cancer based on the location of the target lesion [12,13,14]. Site-specific thresholds avoid the decrease in specificity that occurs when using the SUV threshold of 2.5. SUV can also be compromised by variability between and within individuals using the same imaging device; serial examinations of a patient using different devices adds to inconsistency and reduces accuracy. Other attempts have been reported to standardize the SUV but have varying levels of accuracy and are not widely used [15,16,17].

In this report, we examined the utility of exploiting the remarkable stability of the liver as a visceral reference to create a novel SUV normalization formula [18,19,20]. First, a normal liver SUV_max_ constant (k) was derived from weighted meta-analysis of published studies using calibrated hardware and dosing procedures; we employed “k” in our formula in an effort to enhance the predictive value of the SUV_max_ [7,21]. Next, we used different anatomic site-specific SUV thresholds to discriminate malignant from non-malignant lesions and added this to our algorithm to further optimize SUV_max_. Finally, the formula for optimized SUV_max_ was compared with conventional SUV_max_.

## 2. Materials and Methods

### 2.1. Patients

FDG PET-CT scans and matched histopathologic reports were available for a total of 393 patients studied between 2005 and 2012; approximately 1000 other patients without available pathology data were, of necessity, excluded.

Data were stratified into three groups according to anatomic sites of possible cancer:(1)Mediastinal and/or hilar lymph nodes (MHLNs) in patients with known lung carcinoma at staging or restaging of the disease (*n* = 154, 105 initial staging and 49 restaging).(2)Extra-thoracic lymph nodes (ETLNs) (*n* = 143) for patients with malignant lymphoma (*n* = 48 patients), head and neck cancer (*n* = 21), breast cancer (*n* = 18), esophageal cancer (*n* = 11), colorectal carcinoma (*n* = 24), ovarian-uterine cancer (*n* = 10), and pancreatic cancer (*n* = 11) (all patients initial staging).(3)Hepatic parenchyma in patients with known colorectal cancer (*n* = 96 patients, 73 initial staging and 23 restaging).

Image interpretation, performed by a single nuclear medicine specialist (RB), was based on site-specific and liver-corrected SUV_max_ calculation (optSUV_max_), as well as conventional SUV_max_ (conSUV_max_) target lesion assessments; the interpretation was blinded to the pathology findings. Subsequently, accuracy was determined by comparison with the gold standard of biopsy pathology of the target lesion according to the patient reports. Multiple image scanners were used, including GE Discovery 600 PET-CT (*n* = 178 patients), Philips Gemini GXL PET-CT (*n* = 118), and Siemens Biograph 16 slice high resolution PET-CT (*n* = 97).

For each patient, the following data were recorded at the time of scanning: age in years, weight in kilograms, gender, type of suspected/proven malignancy, and imaging scanner. Scan-related data included serum glucose at the time of FDG administration, total activity of injected FDG, time from FDG injection to initiation of image-bed position acquisition, and SUV_max_ generation in normal liver parenchyma. No patient with known malignancy had received cytotoxic systemic chemotherapy or radiation therapy within 6 months of examination. 

Patients fasted prior to scanning: overnight fasting for those scheduled for the morning or at least 6 h fasting for those scheduled for the afternoon. Patients were also instructed to avoid strenuous activity for at least 24 h prior to the examination and encouraged to drink plain, unflavored water; compliance with these instructions was not verified.

Intravenous injection of the FDG radiopharmaceutical was typically accomplished using venous access, avoiding indwelling ports when possible. Scan acquisition usually began 55–100 min after glucose administration to ensure relatively constant liver activity as described by Laffon et al. [19]. The final administered dose of FDG was calculated for each patient by measurement of FDG activity in the syringe prior to and following injection. 

Image acquisition included low-dose CT scan to allow for anatomic correlation and attenuation correction of the PET data per the manufacturer’s instructions. To avoid the potential for reconstruction artifacts, no patient received oral or intravenous contrast for the CT portion of the examination [8,22]. PET acquisition was initiated immediately after the CT scan, with all patients scanned from the skull base to the mid-thigh; those with known head and neck cancer also had inclusion of the skull vertex. 

Calibration procedures were followed for the PET scanner and the dose calibrator using check values performed on a weekly basis [21]. Quality control procedures utilized a germanium (^68^Ge) source and the manufacturers’ suggested protocol. Vendor-recommended processing was followed for each examination, including filtered back-projection for the CT scans; PET emission data were subjected to iterative reconstruction processing. All FDG PET-CT studies were interpreted on a HERMES Nuclear Diagnostics™ workstation utilizing Volume Display software. 

### 2.2. ConSUV_max_ Calculation

The calculated conventional SUV_max_ included the patient’s body weight, uptake of glucose within the target lesion, and the injected dosage of FDG based on the following calculation:SUV_max_ − bw [g/mL] = T × bw × 1000/D
with T representing the maximum voxel reading for the radioactivity concentration in the target lesion in becquerels per milliliter (Bq/mL), D representing the applied dose (in Bq) of the radiopharmaceutical at the time of image correction, and bw reflecting that the SUV was normalized to the patient’s body weight in kilograms [23]. The liver SUV was obtained by identifying a circular region of interest in the liver parenchyma with three successive measurements from a single region in the coronal plane; the region of interest encompassed the entire abnormality and was invariably 1 cm in diameter or greater. The mean of the three SUV_max_ values in the hepatic parenchyma was recorded. Areas of increased uptake within the liver, such as metastases or primary liver malignancy, were avoided, as were photopenic defects attributed to previous therapeutic intervention or cyst formation. No patient had significant liver disease precluding the procurement of the reference normal hepatic parenchymal SUV.

### 2.3. OptSUV_max_ Calculation 

The target lesion encompassed the entire region of abnormal uptake to ensure that a maximal uptake value was utilized for the purposes of comparison to liver SUV. The target lesion result was normalized with the SUV_max_ obtained from normal liver, an approach that minimized risk of variability and potentially falsely low values that might alter the interpretation. Optimal SUV_max_ (optSUV_max)_ was obtained from the value derived from the target lesion (T) that was modified by the patient’s normal liver SUV_max_ and a proprietary liver constant k derived from a weighted meta-analysis of the literature [7,24,25,26,27,28,29]. L is normal liver SUVmax. This remodeled SUV_max_ abbreviated as optSUV_max_, is shown by the following formula: 

optSUV_max_ = (T)(*k*)/L

Because the modifications are essentially those involving fine-tuning of the liver SUV_max_, the final equation is depicted as follows:optSUV_max_ = T/L_mod_

### 2.4. Statistical Methods

Two datasets were generated: conSUV_max_ and optSUV_max_. For conSUV_max_, the definition of malignancy has been reported by others [11,12,14,30]. SUV_max_ findings were categorized as “indeterminate” based on a 20% decrease from the SUV_max_ threshold and “negative” for malignancy if the value was greater than a 20% decrease from the SUV_max_ threshold value. These additional threshold values for defining malignancy were previously established in 68 patients undergoing staging for primary lung carcinoma [14,30]. In the optSUV_max_, the patient-derived normal liver parenchymal SUV_max_ and its adjustment using a constant “k” from a review of the peer-reviewed literature was employed.

For the optSUV_max_**,** anatomic site-specific SUV_max_ thresholds defined the optSUV_max_ as positive, negative, or indeterminate: MHLN thresholds: SUV ≥ 4.0 for defining the presence of cancer, 3.2–3.9 for indeterminate classification, and ≤3.1 for a negative classification (not fulfilling criteria for cancer).ETLN thresholds: SUV ≥ 2.5 for defining the presence of cancer, 2.1–2.4 for indeterminate classification, and ≤2.0 as negative for cancer.Hepatic parenchymal thresholds: SUV ≥ 4.0 and a TLR (target lesion to normal liver SUV_max_ ratio) > 2.0 defining the presence of cancer; a value of 3.2–3.94 and a TLR of 1.7–1.9 for indeterminate classification, and an SUV in the target lesion of ≤3.1 and TLR ≤ 1.6 for excluding the presence of cancer.

All statistical comparisons used the Student’s *t*-test at a significance level of 0.05.

## 3. Results

### 3.1. Patient Characteristics

Patient population characteristics are summarized in Table 1 for the three different patient groups according to site. A total of 207 males and 186 females were investigated, with subject ages ranging from 28 to 84 years. Only patients with confirmed histology were included in the analyses.

### 3.2. Group 1: MHLN Lung Cancer Cohort

The sensitivity and specificity for optSUV_max_ was 83.8% and 88.7%, respectively, with a positive predictive value (PPV) of 83.8% and a negative predictive value (NPV) of 88.7% (Figure 1). The sensitivity, specificity, PPV, and NPV for conSUV_max_ were 80.7%, 96%, 30.1%, and 50%, respectively.

The results of optSUVmax and conSUVmax classified as indeterminate were further analyzed. Their effects on sensitivity, specificity, PPV, and NPV when grouped with positive or negative findings are shown in Figure 1. 

### 3.3. Group 2: ETLN Cohort

The sensitivity and specificity for optSUV_max_ were 92.1% and 80.0%, respectively. optSUV_max_ had a PPV of 82.4% and an NPV of 90.1% (Figure 2). The sensitivity, specificity, PPV, and NPV for conSUV_max_ was 77.8%, 27.6%, 40%, and 66.7%, respectively.

The results of optSUV_max_ and conSUV_max_ classified as indeterminate were further analyzed. Their effects on sensitivity, specificity, PPV, and NPV when indeterminate cases were grouped with positive or negative findings are shown in Figure 2. 

### 3.4. Group 3: Hepatic Parenchymal Colorectal Carcinoma Cohort 

Calculated sensitivity and specificity for the optimized SUV_max_ positive and negative categories were 96.1% and 88.8%, respectively, with a corresponding PPV of 96.1% and NPV of 88.8% (Figure 3). The sensitivity, specificity, PPV, and NPV for conSUV_max_ was 82.3%, 23.0%, 55.3%, and 50.0%, respectively.

The results of optSUV_max_ and conSUV_max_ classified as indeterminate were further analyzed. Their effects on sensitivity, specificity, PPV, and NPV when indeterminate cases were grouped with positive or negative findings are shown in Figure 3.

### 3.5. Total Cohort

Of 393 patients reviewed, 181 (46%) had a change in classification based on the use of optSUV_max_ (Table 2). Of the conSUV_max_ determinations that were classified as malignant (+), 93% were reclassified as non-malignant (−) when the optSUV_max_ algorithm was used, and all were confirmed as non-malignant by pathology results. Three patients with conSUV_max_ negative determination were reclassified as positive by optSUV_max_, and all three had pathologic documentation of malignancy. The significant differences defined in the three patient cohorts for the indeterminate classification were maintained when combining all three groups. The combined results of the three cohort groups are shown in Table 2 and Figure 4. Sensitivity and specificity for optSUV_max_ positive and negative categories were 91.4% and 85.1%, respectively, with corresponding PPV of 87.7% and NPV of 89.6%, significantly higher than those of the conSUV_max_ (conSUV_max_ sensitivity and specificity findings of 80.3% and 20.5%, respectively). OptSUV_max_ increased PPV and NPV from 38.6% to 87.7% and 57.0% to 89.6%, respectively, as shown in Figure 4.

Figure 5 illustrates multiple cases of FDG PET scans with discrepancies in interpretation between conSUV_max_ and optSUV_max_.

## 4. Discussion

There is a critical need for standardization of interpretation of glucose uptake with FDG PET imaging [15], and the lack of reproducibility has called into question the validity of the SUV results [31,32]. Multiple publications have addressed the issue of potential sources of error when implementing SUV as a semi-quantitative indicator of the degree of glucose metabolism within a region of interest [33,34]. SUV interpretation and reporting is fraught with inconsistency and variation, resulting in only modest acceptance by the referral community [17]. In contrast to SUV, more precise quantitative methods—including compartmental modeling, spectral analysis, and graphical methodology—have limited applicability outside of academic center environments owing to significant technical demands, patient inconvenience, and limited patient tolerance [17]. 

The Society of Nuclear Medicine Procedural Guidelines indicate that normal liver parenchyma generates an SUV of 2.0–3.0 [21], consistent with the quantitative SUV calculation for normal liver parenchyma of approximately 2.0, according to Ramos et al. [7]. In a small survey of healthy male subjects, liver uptake was consistently high and homogeneous with a low coefficient of variation between sequential scans [35]. The accuracy of the SUV generated in the normal liver was not altered by correction for serum glucose or normalization for body surface area or lean body mass [27], although calculations for assessing the response to therapy emphasized the need for the application of the SUL (standardized uptake value corrected for lean body mass) in the PERCIST criteria proposed by Wald et al. [20]. 

SUV interpretation to assess malignancy requires no additional patient effort and is easily generated at the time of image reconstruction and reporting. The SUV-to-background ratio has supplanted tumor-to-background ratio in most instances [23]. A variety of tissues have been proposed in the literature for background comparison, including blood pool activity, cerebellum, muscle, mediastinal blood pool, lung, and normal liver [34]. The liver SUV is uniquely constant among tissues (+/−2.5%) over the course of the initial 50–110 min following FDG injection, but constancy declines after this time interval. [19]. Given that only the liver SUV remains relatively stable over time as opposed to other tissues [18], the use of the liver SUV appears to be ideal for background comparison [20].

Mean liver SUV has been proposed rather than maximal SUV for the purposes of comparative analysis with other tissues [20]. However, we preferred use of SUV_max_ for the target lesion and the liver in this study owing to potential variation of the mean when defining a region of interest within an imaging abnormality or around the entire apparent metabolic aberration. 

The importance of this concept should be clear in the current imaging environment, where patients may have sequential FDG PET studies at the same location with service provided by coach carriers that may bring different devices to the same location or different coach carriers with different vendor machines, or where patients may procure studies from more than one fixed site with the variances in manufacturers’ equipment. 

To our knowledge, no work to date has utilized the patient’s normal liver SUV_max_ that was adjusted by a constant factor derived from meta-analysis of the literature to precisely define the normal liver SUV. The SUV max was then optimized (optSUV_max_) by incorporating anatomic site-specific SUV threshold criteria. 

Our retrospective study has multiple limitations. First, the number of patients in each study cohort was relatively small, although the differences between optSUV_max_ and conSUV_max_ results were sufficiently wide to suggest that the population size was sufficient to generate valid findings. Second, interpretation was undertaken by a single nuclear medicine physician, an approach that limits inter-observer variability but raises concerns about intra-observer variability, internal bias, and generalizability of the findings for widespread use by others. Third, central pathology review was not undertaken, raising the possibility of ascertainment bias. Fourth, the derived algorithm needs to be prospectively validated for utility in diagnosis of treatment response, including metabolic stability, metabolic progression, or complete or partial metabolic response utilizing EORTC criteria [36] or PERCIST criteria [20,37]. Finally, this retrospective analysis is also potentially subject to patient selection bias, including bias arising from inclusion of only those with matched pathology reports [38]. We chose to focus on SUVmax rather than SUVpeak, although others may find the latter preferable [39]; fortunately, SUVmax is unaffected by steatosis. [40] Despite these potential shortcomings, the thresholds used were obtained from previously defined and specified quantitative values, which provides support for the results being reported in this validation dataset. 

Our patients with optimized SUV_max_ readings falling into an indeterminate classification are between the extremes associated with high and low likelihoods of malignancy, with respective values of >80% and <20%. Those individuals with intermediate-to-just-below-threshold optSUV_max_ readings may require biopsies, whereas the patients at the opposite end of the spectrum with low probabilities of malignancy may be appropriately placed into an active objectified surveillance (AOS) category with short-term re-evaluation to ensure stability. 

Reproducibility and validation of ^18^F-FDG PET algorithms are still major challenges, according to a recent review by Anan [41]. He notes that the absence of unanimously recognized reference values and definitions has hampered clinical use of ^18^F-FDG PET, compounded by a lack of uniformity of the image processing platforms required to analyze features. Manipulation and assessment of a single image set in two different software platforms may result in dissimilar feature values, and harmonization efforts are complicated. Variations of imaging procedure, ^18^F-FDG activity, image reconstruction, data comprehension, and uptake time are significant. The situation can be solved by standardization of the radiomic features definition and quantification.

By adding the capability of an objective quantitative answer to the clinician’s question of what is malignant and what is not, confidence in the procedure from both the interpreting and referring physicians’ perspective will likely escalate, with the objective findings facilitating optimal patient management [42]. The methods we report herein can be reliably accomplished on any PET imaging device with available software to enable generation of the SUV. There is no need for correction for lean body mass, body surface area, or serum glucose level. 

## 5. Conclusions

Current methods of FDG PET-CT interpretation for malignancy are prone to variability and inaccuracy. We present a method—optimized SUV_max_—that enhanced accuracy when compared with conventional SUV_max_ interpretation. This was achieved by (1) normalizing the target lesion SUV_max_ (T) based on the patient’s normal liver SUV_max_ (L); (2) including a constant k derived from a meta-analysis review of published SUV data of normal liver SUV_max_ to optimize L; and (3) adjusting the method’s formula to account for SUV_max_ thresholds of specific anatomic sites such as mediastinal and/or hilar lymph nodes (MHLNs), extra-thoracic lymph nodes (ETLNs), and hepatic parenchyma. When compared with conventional SUV_max_ in a series of 393 patients, optimized SUV_max_ significantly improved diagnostic sensitivity, specificity, PPV, and NPV.

## 6. Patents

The work herein is contained within the following patents: European Union patent 3,048,977; US patent 10,674,983; and US patent 11,382,586.

## Figures and Tables

**Figure 1 diagnostics-13-01580-f001:**
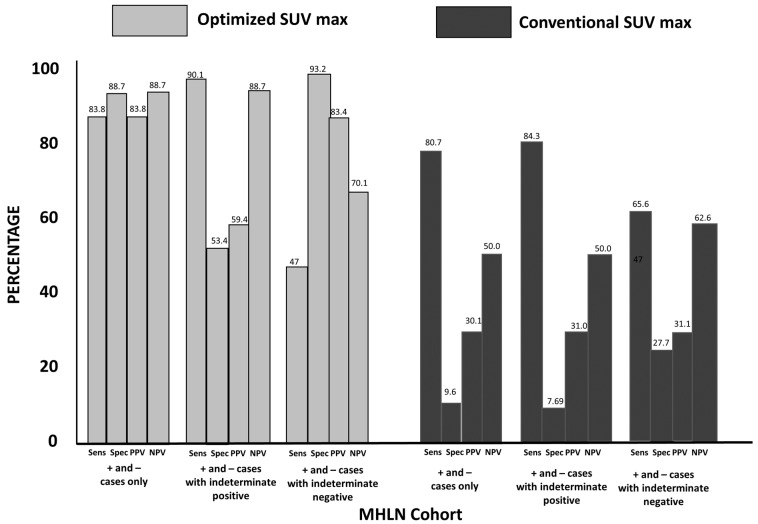
Mediastinal-Hilar Lymph Node (MHLN) Cohort Findings for OptSUVmax vs. ConSUVmax: The comparative sensitivities for cases (excluding indeterminate) for optSUV_max_ (left graph (light shading), far left bar) vs. conSUV_max_ (right graph (dark shading); far left bar) were 83.8% vs. 80.7%, respectively (*p* = 0.36). Markedly different specificities for cases (excluding indeterminate) were 88.7% (left graph; bar second from the left) vs. 9.6% (right graph; second bar from the left) for optSUV_max_ vs. conSUVmax, respectively (*p* < 0.0001). The comparative accuracies for cases (excluding indeterminate) were 86.7% vs. 29.5% for optSUV_max_ vs. conSUV_max_, respectively. When indeterminate findings were grouped with positive for malignancy (set of four middle bars in the light-shaded graph on the left for optSUV_max_ and set of four middle bars in the dark-shaded graph on the right for conSUV_max_), sensitivity increased but specificity declined greatly; when indeterminate findings were grouped with negative for malignancy (set of four right bars in the light-shaded graph on the left for optSUV_max_ and set of four right bars in the dark-shaded graph on the right for conSUV_max_), sensitivity declined greatly, indicating a greater risk for false-negative interpretation.

**Figure 2 diagnostics-13-01580-f002:**
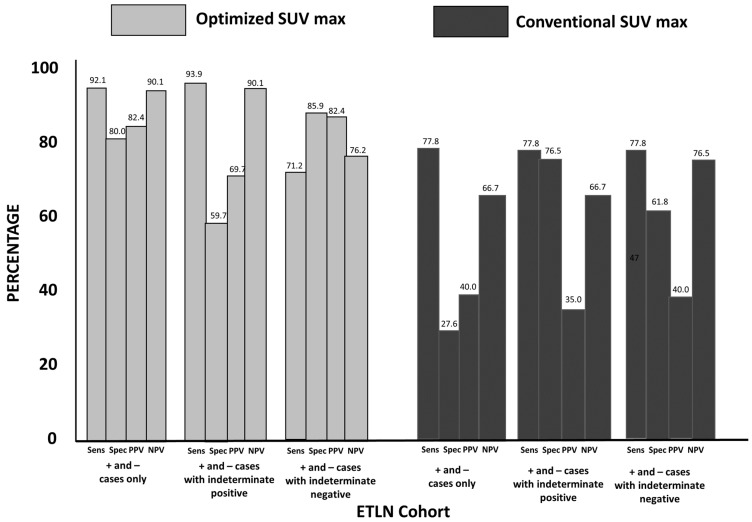
Extra-Thoracic Lymph Node (ETLN) Cohort Statistical Findings for OptSUVmax vs. ConSUVmax (see description in Figure 1): The sensitivity for optSUV_max_ vs. conSUV_max_ was 92.1% vs. 77.8%, respectively (*p* = 0.043). The specificity for optSUV_max_ vs. conSUV_max_ was 80.1% vs. 27.6%, respectively (*p* < 0.0001). The optSUV_max_ vs. conSUV_max_ accuracy was 87% vs. 46.8%, respectively (*p* < 0.0001). When optSUV_max_ indeterminate findings were grouped with positive for malignancy, specificity declined, indicating a greater risk for false-positive interpretation.

**Figure 3 diagnostics-13-01580-f003:**
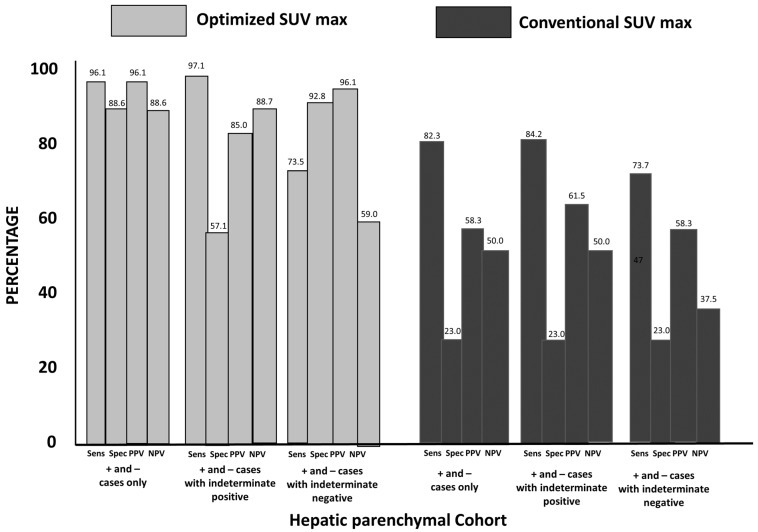
Hepatic Parenchymal Cohort Statistical Findings for OptSUVmax vs. ConSUVmax (see description in Figure 1): optSUV_max_ sensitivity and specificity were 96.1% and 88.8%, respectively, with a PPV of 96.1% and an NPV of 88.8% for positive or negative FDG PET-CT determinations relating to hepatic metastases. The sensitivity, specificity, PPV, and NPV for conSUV_max_ were 82.3%, 23.0%, 55.3%, and 50.0%, respectively, for positive and negative cases. When indeterminate cases were grouped as positive or negative, the results with optSUVmax were superior compared to conSUV_max_ determinations.

**Figure 4 diagnostics-13-01580-f004:**
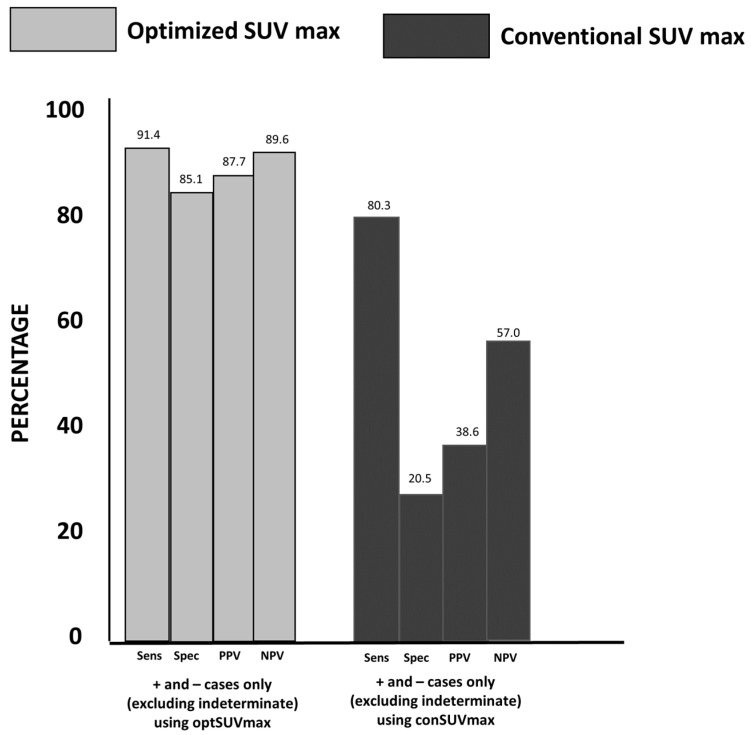
Changes in Sensitivity, Specificity, PPV, and NPV between optSUV_max_ and conSUV_max_ for all cohorts combined (see description in Figure 1). Optimized SUVmax improved the accuracy of FDG PET-CT determinations when compared with conSUVmax in discriminating between patients with malignant vs. non-malignant pathologic outcomes.

**Figure 5 diagnostics-13-01580-f005:**
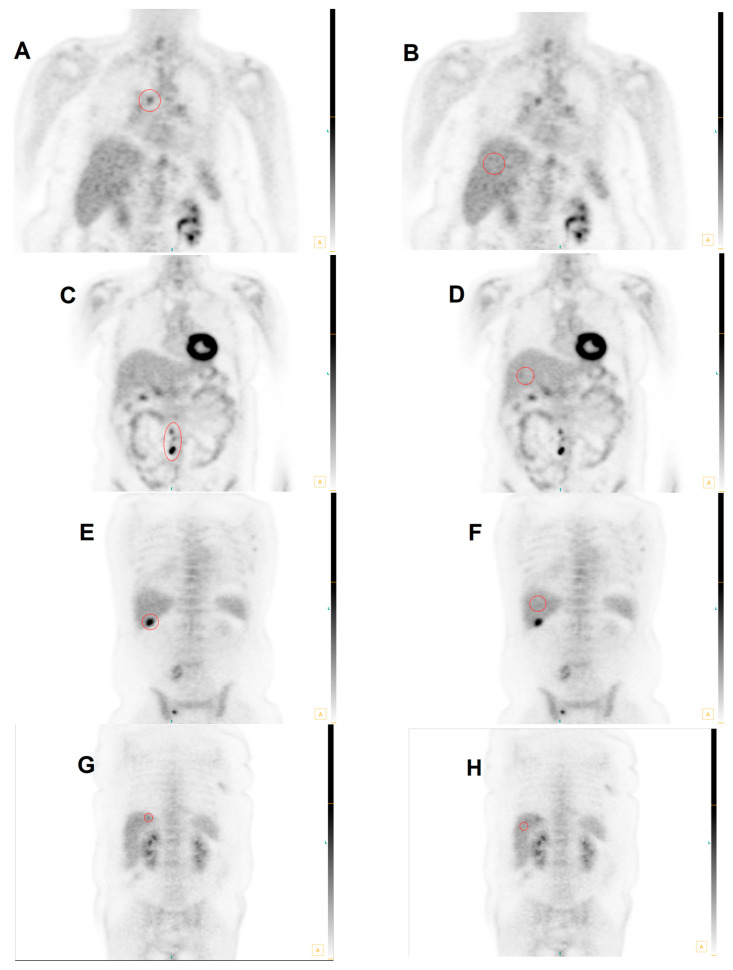
Four Cases Illustrating Differences between conSUV_max_ and optSUV_max_. Case 1: False-positive conSUV_max_ in MHLNs. In (**A**), the mediastinal abnormality is defined in the red circled region of interest and produces a conSUV_max_ of 9.80, so this is considered to be positive for malignancy. (**B**) shows the red circled region to produce the conSUV_max_ liver reference in normal liver parenchyma as 7.84; therefore, the formula-calculated optSUVmax is 2.5. The site-specific threshold for defining malignancy in the MHLNs is SUV ≥ 4.0, so optSUV_max_ is considered to be negative for malignancy. Pathology confirmed the absence of cancer at this site. Case 2: False-negative conSUV_max_ in ETLNs: In (**C**), the right para-aortic nodes depicted in the region of interest demarcated by the red circle have a conSUVmax of 1.88, so this is considered to be negative for malignancy. (**D**) shows the conSUVmax liver reference via the encircled red region of interest in normal liver parenchyma to be 0.50 L; therefore, the calculated optSUVmax is 7.5. The site-specific threshold for defining malignancy in the ETLNs is ≥2.5, so the corrected optSUVmax is considered to be positive for malignancy. Pathology confirmed the presence of lymphoma at this site. Case 3: This case represents a false-negative conSUV_max_ in the liver: In (**E**), the abnormality demonstrated in the red circular region of interest of the liver lesion conSUV_max_ is 2.77, so this is considered to be negative for malignancy. (**F**) shows the region defined by the manually drawn conSUVmax liver reference to be 1.02; therefore, the calculated optSUV_max_ is 5.4, with a lesion-to-liver background ratio of >2.0 The site-specific threshold for defining malignancy in the liver is SUV ≥ 4.0, so optSUV_max_ is considered to be positive for malignancy. Pathology confirmed the presence of cancer at this site. Case 4: False-positive conSUV_max_ in the liver: In (**G**), the chosen region of the liver parenchymal abnormality produces a conSUV_max_ of 7.04, so this is considered to be positive for malignancy. (**H**) reveals the conSUVmax liver reference to be 4.88 via the chosen normal parenchymal region delineated in the encircled red region; therefore, the calculated optSUV_max_ is 2.9 with a lesion-to-liver background ratio of < 2.0. The site-specific threshold for defining malignancy in the liver is SUV ≥ 4.0, so optSUV_max_ is considered to be negative for malignancy. Pathology confirmed the absence of cancer at this site.

**Table 1 diagnostics-13-01580-t001:** Patient Demographics, Tumor Types, Normal Liver SUVs. A total of 393 patients constituted the entire study population. The primary malignancy types with numbers are shown for the three anatomic sites: mediastinal and/or hilar lymph nodes (MHLNs), extra-thoracic lymph nodes (ETLNs), and liver. Normal hepatic parenchyma SUV_max_ values are shown for each group’s anatomic site. The threshold or range to ascribe positive, negative, or indeterminate categorization of FDG PET-CT findings is shown. Data are means ± SD (range).

Anatomic Site	MHLN	ETLN	LIVER	Total Group
N (M/F)	154 (86/68)	143 (70/73)	96 (51/45)	393 (207/186)
Age Range (y)	39–84	28–83	32–82	28–84
Cancer Types	Lung cancer (154: 105 initial staging & 49 restaging)	Lymphoma (48) Head & Neck (10) Breast (18) Esophageal (11) Colorectal (24) Ovarian-Uterine (10) Pancreatic (11) (all initial staging)	Colorectal (96: 73 initial staging & 23 restaging)	
Non-disease liver parenchymal SUV	2.72 ± 0.76 (0.9–4.9)	2.42 ± 0.64 (1.1–4.1)	2.50 ± 0.70 (1.2–5.1)	2.54 ± 0.70 (0.9–5.1)
Within-person Coefficient of Variation for normal liver parenchymal SUVs	0.035 (0.0–0.10)	0.037 (0.0–0.20)	0.046 (0.0–0.20)	0.039 (0.0–0.20)
Threshold Values Based on Literature			***	
Positive	≥4.0	≥2.5	≥4.0 & TLR >2.0	
Indeterminate	3.2–3.9	2.1–2.4	3.2–3.9 & TLR ≈2.0	
Negative	≤3.1	≤2.0	≤3.1 & TLR <2.0	

*** TLR = SUV_max_ liver lesion: normal liver SUV_max._

**Table 2 diagnostics-13-01580-t002:** Diagnostic Changes in Interpretation Based on Disparities Between conSUVmax and optSUVmax Results: Data are shown only for patients with changes in the FDG PET-CT determinations of positive vs. negative vs. indeterminate for malignancy. Data are not shown for patients who had no change in FDG PET-CT imaging assignments with conSUV_max_ compared to optSUV_max_. When the optSUV_max_ findings were positive (abnormal), the pathology confirmed this 100% of the time, with no false positives. When the optSUV_max_ was negative (66 patients), the pathology was negative in 60 patients (90.1%).

conSUV_max_	⇢	optSUV_max_	Pathology +	Pathology −
negative		positive	3 (100%)	0
positive		negative	3 (7%)	42 (93%)
negative		indeterminate	9 (36%)	16 (64%)
positive		indeterminate	46 (56%)	36 (44%)
indeterminate		negative	3 (14%)	18 (86%)
indeterminate		positive	5 (100%)	0

## Data Availability

The data presented in this study are available on request from the corresponding author. The data are not publicly available due to privacy concerns.

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
