# Peer review of "Optimized 18F-FDG PET-CT Method to Improve Accuracy of Diagnosis of Metastatic Cancer"

_diagnostics, 2023, doi:10.3390/diagnostics13091580_

Round 1

Reviewer 1 Report

The authors propose an optimized image interpretation method based on a novel SUV normalization formula that uses a liver SUVmax constant and site-especific SUV thresholds to discriminate malignancy. The proposed method enhaces lesion discrimination so it contributes to cancer diagnosis.

The proposed method is simple and applicable but lack of novelty and academic significance and soundness. FDG PET-CT imaging is used as a preliminary diagnostic tool, in order to make a more precise diagnose there are improved tools that allow a more accurate detection of molecular changes that are similarly available. Liver parenchyma glucose uptake can be affected by multiple factors so inclusion of individuals with different pathological comorbidities (diabetes, hypertension or any metabolic syndrome) should be included in the study as controls.

The manuscript is clear well-presented but most recent literature review is recommended, the referenced literature goes from 1956 to 2019. Proposal of an improved image interpretation method is pertinent for the field but the sole comparison of the conventional and the optimized methods has no scientific relevance.

It is recommended to include in the experimental design some borderline pathological conditions in order to test the accuracy of the method under extreme conditions and also consider the variability due to threshold regions and ROI delimitation.

In materials and methods section line 72 the authors state that “data were stratified into three groups according to anatomic sites of possible cancer” explain what was the discriminatory condition used to include 24 colorectal cancer patients in the ETLN group instead of in the liver group

In order to make the images more understandable, it is recommended to improve the figure captions so that they are more descriptive. In the figure caption of all the figures, the authors mention an accuracy value but they do not explain how it was calculated.

Author Response

We appreciate the careful review of Reviewer #1, and have the following modifications and responses:

Paragraph 2: “… diabetes, hypertension and metabolic syndrome…”

Our algorithm does not correct for serum glucose levels for patients with diabetes, hypertension, metabolic syndrome, and other conditions (excluding cystic liver disease and cancer) as such conditions do not affect normal liver SUV calculations (see our Reference 27, Paquet et al, 2004).

Paragraph #3: “Literature review…”

We have added the following 2022 article to the manuscript: Anan N, et al. A review on advances in (18)F-FDG PET/CT radiomics standardization and application in lung disease management. Insights Imaging 2022; 13: 22.

Paragraph #3: “Comparison of the conventional and optimized methods has no scientific relevance.”

We respectfully disagree; this comparison is the essence of our submitted report.

Paragraph #4: “…some borderline pathological conditions…”

We agree that such inclusion would provide useful data to address accuracy; however, this was outside the bounds of our study and we did not have access to such patient cases.

Paragraph #4: Variability due to threshold regions and ROI delimitation—speaks to heterogeneity within the liver.

Higher sensitivity detectors in machines decrease variability—vendors have improved machines, and this creates more uniform SUV. Our algorithm is normalized and does not appear to be significantly influenced by heterogeneity in the normal liver.

Paragraph #5: “…colorectal…”

Colorectal cohort included involvement of peripheral lymph nodes with no liver involvement; these patients did not have liver parenchymal disease. This is separate from those with liver metastases.

Paragraph #6: “…improve the figure captions so that they are more descriptive...”

            We appreciate the opportunity to revise the captions to heighten the readability and comprehensibility. The captions have been expanded accordingly, especially for Figures 1 and 5 (changes to Figure 5 caption included below in RED).

Figure 5. Four Cases Illustrating Differences between conSUVmax and optSUVmax. Case 1: False Positive conSUVmax in MHLNs. In A, the mediastinal abonormality is defined in the red circled region of interest produces a conSUVmax is 9.80, so this is considered to be positive for malignancy. B shows the red circled region to produce the conSUVmax liver reference in normal liver parenchyma to be 7.84; therefore, the formula calculated optSUVmax is 2.5. The site-specific threshold for defining malignancy in the MHLNs is SUV > 4.0, so optSUVmax is considered to be negative for malignancy. Pathology confirmed the absence of cancer at this site. Case 2: False negative conSUVmax in ETLNs: In C, the right para-aortic nodes depicted in region of intetest demarcated by the red circle have a conSUVmax of 1.88, so this is considered to be negative for malignancy. D shows the conSUVmax liver reference via the encircled red region of interest in normal liver parenchyma to be 0.50l therefore, the calculated optSUVmax is 7.5. The site-specific threshold for defining malignancy in the ETLNs is ≥ 2.5, so the corrected optSUVmax is considered to be positive for malignancy. Pathology confirmed the presence of lymphoma at this site. Case 3: This case represents a false negative conSUVmax in the liver: In E, the abnormality demonstrated in the red circular region of interest of the liver lesion conSUVmax is 2.77, so this is considered to negative for malignancy. F shows the region defined by the manually drawn conSUVmax liver reference to be 1.02; therefore, the calculated optSUVmax  is 5.4, with a lesion-to-liver background ratio of  > 2.0  The site-specific threshold for defining malignancy in the liver is SUV > 4.0, so optSUVmax is considered to be positive for malignancy. Pathology confirmed the presence of cancer at this site. Case 4: False positive conSUVmax in the liver: In G, the chosen region of the liver parenchymal abnormality produces a conSUVmax of 7.04, so this is considered to be positive for malignancy. H reveals the conSUVmax liver reference to be 4.88 via the chosen normal parenchymal region delineated in the encircled red region; therefore, the calculated optSUVmax is 2.9  with a lesion-to-liver background ratio of  < 2.0  The site-specific threshold for defining malignancy in the liver is SUV > 4.0, so optSUVmax is considered to be negative for malignancy. Pathology confirmed the absence of cancer at this site.

Paragraph #6: Accuracy calculation:

We used the following definition (Accuracy = TP + TN / TP + TN + FP + FN)

Reviewer 2 Report

This study aimed to compare the accuracy of a new suggested normalized SUV measurement method with conventional SUV reading in the diagnosis of malignancy. Histopathology was served as the gold standard. The manuscript is well written. The study has a few major limitations that are all described in the discussion. Another limitation which is not mentioned is the handling of the "intermediate" readings. This group is quite large and add these to the positive or the negative findings may lead to bias. Moreover, excluding these "intermediates" may also lead to uncorrected results. 

Normalization and standardization of SUV measurements is very important, especially when comparing serial studies. The suggested tool might be of value. However other tools such as SUVpeak should be mentioned and discussed.

Specific comment:

Page 3 Line 122: “concentration of glucose” should be dropped as this formula does not adjust SUV for glucose concentration.

Page 3 Line 125: Although technically correct, this equation is ambiguous in its application and should be written as SUVmax-bw = T×bw×1000/D

Page 3 Line 126: “with T representing radioactivity concentration in the target lesion” should read “with T representing the maximum voxel reading for the radioactivity concentration in the target lesion”

Page 3 Line 129: “circular region of interest” Approximately what size?

Page 3 Line 130 “three successive measurements in the coronal plane” Are these three circles beside each other or three circular ROIs on successive coronal planes?

Page 3 Line 146: L is not defined. Is it the “patient’s normal liver SUVmax”?

Author Response

We appreciate the careful review by Reviewer #2, and have the following modifications and responses:

Paragraph #2: “…SUVpeak”

As SUVpeak is not used as commonly as SUVmax, we have chosen not to include it. However, the following article has been added as a reference, and text has been added to the Discussion:

Vanderhoek M, et al. Impact of the definition of peak standardized uptake value on quantification of treatment response. J Nucl Med 2012; 53: 4-11.

In the Discussion: “We chose to focus on SUVmax rather than SUVpeak, although others may find the latter preferable [39].”

Page 3 Line 122: “concentration of glucose” should be dropped as this formula does not adjust SUV for glucose concentration.

The reviewer is correct. We have changed the wording from “concentration of glucose” to “uptake of glucose”

Page 3 Line 125: Although technically correct, this equation is ambiguous in its application and should be written as SUVmax-bw = T×bw×1000/D

We agree and have modified the text accordingly.

Page 3 Line 126: “with T representing radioactivity concentration in the target lesion” should read “with T representing the maximum voxel reading for the radioactivity concentration in the target lesion.”

We agree and have modified the text accordingly.

Page 3 Line 129: “circular region of interest” Approximately what size?

Text has been added for clarification: the region of interest encompassed the entire abnormality, and was invariably 1cm in diameter or greater.  

Page 3 Line 130: “…three successive measurements in the coronal plane” Are these three circles beside each other or three circular ROIs on successive coronal planes?

These are based on a single region in the coronal plane involving sequential slides. This has been added to the text.

Page 3 Line 146: “L is not defined. Is it the “patient’s normal liver SUVmax”?”

            Yes, L is normal liver SUVmax—text is modified accordingly.

Reviewer 3 Report

I congratulate with the authors, the work is well structured, interesting and scientifically valid. I have only a few minor considerations.

Minor questions:

1)      in the methods section (page 3, lines 134-135) the authors state that no patient had significant liver disease such as to preclude the achievement of the reference normal liver parenchymal SUV. I would suggest adding some examples of the liver diseases that have been excluded. For example alcoholic hepatitis, fatty liver disease, viral hepatitis, cholestasis etc. Just a couple of lines.

2)      The authors in Table 1 describe the characteristics of the patients. I would suggest also including (if possible) the mean age for all subgroups considered in the present study.

3)      The authors evaluated whether the patients' optSUVmax data were related to some classical markers such as CA 15-3 for breast cancers, CEA for colorectal cancers, CYFRA 21-1 for lung cancers, or LDH and beta 2 microglobulin for hematologic malignancies? It would be an interesting implementation that would enrich the work. of course if the data is available.

Author Response

We appreciate the careful effort by Reviewer #3, and have the following modifications and responses:

Q1: Liver condition that can affect SUVmax:

None of the patients in our study had hepatitis, fatty liver, or cirrhosis. Cholelithiasis does not apparently affect SUVmax (or the algorithm that we describe) according to Abele et al (new article included—reference 40), although disparate results have been reported.

Q2:  Mean age

Mean was not assessed (only range is provided), but all are 18 years of age

Q3: Correlation of optSUVmax with other biomarkers (e.g., CA 15-3, CEA, CYFRA 21-1, etc.)

This is an excellent idea that was not feasible in this first study. However, we will include in future studies.